# A Community-Based Intervention for Improving Medication Adherence for Elderly Patients with Hypertension in Korea

**DOI:** 10.3390/ijerph16050721

**Published:** 2019-02-28

**Authors:** Kang-Ju Son, Hyo-Rim Son, Bohyeun Park, Hee-Ja Kim, Chun-Bae Kim

**Affiliations:** 1Research Institute for Healthcare Policy, Korean Medical Association, Seoul 04373, Korea; sonkangju@hanmail.net; 2Hongcheon County Hypertension and Diabetes Registration and Education Center, Kangwon Province, Hongcheon 25135, Korea; hc_health@naver.com; 3Hongcheon County Health Center, Kangwon Province, Hongcheon 25135, Korea; pbh0118@korea.kr; 4Hoengseong County Health Center, Kangwon Province, Hoengseong 25220, Korea; Khja1231@korea.kr; 5Department of Preventive Medicine, Yonsei University Wonju College of Medicine, Wonju 26426, Korea; 6Institute for Poverty Alleviation and International Development, Yonsei University, Wonju 26493, Korea

**Keywords:** hypertension, medication adherence, community-based intervention, difference-in-difference regression

## Abstract

The chronic disease management program, a community-based intervention including patient education, recall and remind service, and reduction of out-of-pocket payment, was implemented in 2005 in Korea to improve patients’ adherence for antihypertensive medications. This study aimed to assess the effect of a community-based hypertension intervention intended to enhance patient adherence to prescribed medications. This study applied a non-equivalent control group design using the Korean National Health Insurance Big Data. Hongcheon County has been continuously implementing the intervention program since 2012. This study involved a cohort of patients with hypertension aged >65 and <85 years, among residents who lived in the study area for five years (between 2010 and 2014). The final number of subjects was 2685 in both the intervention and control region. The indirect indicators were analyzed as patients’ adherence and level of continuous treatment using the difference-in-difference regression. The proportion of hypertensive patients who continuously received insurance benefits for >240 days in 2014 was 81.0% in the intervention region and 79.7% in the control region. The number of dispensations per prescription and the dispensation days per hypertensive patient in the intervention region increased by approximately 10.88% and 2.2 days on average by month, respectively, compared to those in the control region. The intervention program encouraged elderly patients with hypertension to receive continuous care. Another research is needed to determine whether further improvement in the continuity of comprehensive care will prevent the progression of cardiovascular diseases.

## 1. Introduction

The United Nations (UN) reached an agreement titled “Political Declaration of the High-Level Meeting of the General Assembly on the Prevention and Control of Non-Communicable Diseases (NCDs)” in 2011 to respond to the crisis of double burden of NCDs [1]. According to various studies such as the Global Burden of Disease (GBD) Study 2010, GBD Study 2013, and GBD Study 2015 [2,3,4,5], NCDs now account for more than one-half of the global burden of disease. Cardiovascular diseases (CVD) account for about one-half of NCD deaths. In addition to health-related behaviors like tobacco smoking, unhealthy diet, and physical inactivity, hypertension and diabetes were also the most common risk factors for CVD [6]. Based on recent mortality data on the cause of death released by the National Statistical Office, not only the mortality rate from cerebrovascular diseases but also the burden of NCD, including CVD, steadily increased in Korea [7,8]. Also, significant health inequalities across NCDs, such as hypertension, diabetes, ischemic heart disease, were observed mainly in Korean women [9]. 

### 1.1. Community-Based Intervention for Chronic Disease Management Program

Community-based interventions for controlling high blood pressure, such as engaging community health workers in hypertension prevention, team-based care to improve blood pressure control, and screening for hypertension, have played a key role in reducing morbidity and mortality from CVD [10]. Of course, the Korean government targeted hypertension and diabetes, the major risk factors for CVD, through the prevention and control program, too. The following community-based interventions were put in place [11]. First, it was recognized to all residents that the community campaign could manage chronic diseases (including hypertension and diabetes) and encourage active treatment after diagnosis. Second, the registration and education program induced early detection of asymptomatic patients with hypertension, self-awareness, and proper care. Third, to strengthen the healthcare system, the government is applying the registration and management programs for controlling hypertension through the chronic care model at the primary care site [12]. 

In 2005, the Daegu Metropolitan City conducted the demonstration program on high-risk groups of CVD. On this occasion, as of 2018, 19 regions in Korea were operating the registration and education center for hypertension and diabetes. The target group of this community-based intervention includes patients with hypertension and diabetes aged >30 years who lived within the place. Primary healthcare institutions and pharmacies that treat hypertension and diabetes in the community also participate as partners of this center. To keep the intervention running, the center is providing a registration fee (about US$ 1 for patients aged ≥65 years and US$ 4 for those aged 30~64 years) for each patient per year to the clinic. All patients, regardless of their age, are notified of their regular care schedule through the recall and remind service when they missed medical care. Moreover, the center provides health education and counseling services on diseases, nutrition, and exercise to improve chronic diseases self-management for patients. Finally, this center provides subsidies (about US$ 3 per month per patient) as out-of-pocket payment only to registered elderly patients aged ≥65 years every time he or she receives medical care utilization after registering to improve medication adherence [13,14].

### 1.2. Approaches for Assessing Community-Based Intervention to Improve Medication Adherence

Although several systematic reviews have shown that some community-based interventions are effective to improve blood pressure control in the community [15,16], few studies have looked at the analysis of interventions to enhance medication adherence and blood pressure control in hypertension. Medication adherence is generally defined as the extent to which patients take drugs or medicine as prescribed by their health care providers [17]. The Korea Centers for Disease Control and Prevention (KCDC) developed indicators of medication adherence for hypertension through the Korean National Health Nutrition Survey (KNHANES) and the Community Health Survey [18]. However, they have the following limitations: (1) These indicators provide only fragmentary information by year; (2) because most indicators are produced by cross-sectional studies, individual trends of medication adherence cannot be identified; (3) memory bias may exist due to a survey; and (4) securing representation in a sample survey may be difficult. Consequently, studies involving longitudinal and time-series data are needed for evaluating a community-based hypertension intervention with patients’ adherence.

Korea has already achieved mandatory universal health coverage since 1989 with the national health insurance system. The National Health Insurance Service (NHIS) developed the Korean National Health Insurance (KNHI) Big Data platform in 2016 [19]. Thus, this KNHI Big Data can identify the actual medical use, which cannot be identified in the survey. In particular, individual and collective medical care utilization can be tracked by establishing an entire population cohort [18]. A study using the KNHI Big Data should be conducted to understand how patients with hypertension change not only their medical care utilization but also medication adherence over time [20,21,22,23].

Therefore, this study aimed to determine whether there was a difference in the medication adherence among elderly patients with hypertension using personalized health information data of the KNHI before and after the community-based intervention in Korea. 

## 2. Materials and Methods 

### 2.1. Research Design and Study Region

This study used a non-equivalent control group design that assessed patients with hypertension in the regions regarding whether the community-based intervention was implemented or not. In addition, at the beginning of the study period, a cohort design that could set up and track the study population was selected. 

The community-based intervention for managing hypertension had been implemented for 6 years is Hongcheon County. This program was launched in July 2012 and is still ongoing. During the early stage, 7 sub-health centers and 2 clinics were only starting. Currently, 7 sub-health centers, 22 clinics, and 17 pharmacies have participated all together as partners. In particular, the registration rate was 100% for elderly hypertensive patients over 65 years old as of 2018. Thus, Hongcheon County was selected as an intervention region [13]. Hoengseong County is geographically close to Hongcheon County and has a very similar population structure as Kangwon Province. Hoengseong County is carrying out the prevention program for CVD to promote the health statuses of residents; however, the community-based intervention was not yet implemented. Thus, Hoengseong County was selected as the control region. 

### 2.2. Data and Study Subjects

Figure 1 shows the process of extracting customized data requested from the NHIS (approval no. NHIS-2018-1-110) for this study. Considering the case of Hongcheon County, it was designated as a research period for every two years, 2010–2014, before and after implementing the program. Study subjects in the intervention and control regions were operationally defined as patients who were prescribed anti-hypertensive drugs after being diagnosed with hypertension. Hypertensive patients were identified according to the ICD-10 codes I10–I15 in the database. Moreover, the anti-hypertensive drug used was classified as 214 (blood pressure reduction drug) based on the No. 196 Operating Procedure (legislation on 2009.8.24) “Regulations on Drugs and Other Classification Numbers” of the Food and Drug Safety Administration [19]. 

In order to assess the effectiveness of this intervention program, patients with similar characteristics in both the intervention and control regions should be selected in order to track and observe them for the same study period. Therefore, this study was limited to elderly patients with hypertension aged over 65 and under 85 years old, among residents who lived in each district for 5 years between 2010 and 2014. Patients aged <85 years were selected in 2010 because the assessment of program effectiveness was considered meaningless as the participation of the super-aging population (aged ≥85 years) was expected to be difficult [24]. 

As of 2010, the population in the intervention region was 21,783 people aged ≤29 years, 34,718 aged 30~64 years, 11,950 aged 64~84 years, and 1002 aged ≥85 years. The total population was 69,453. As of the same year, the population in the control region was 12,919 people aged ≤29 years, 21,686 aged 30~64 years, 8671 aged 64~84 years, and 757 aged ≥85 years. Overall, the total population was 44,033. Regarding the overall population, the intervention region had an approximately 1.58 times higher population size than the control region. According to the current status of the population living continuously for 5 years between 2010 and 2014, based on the eligibility database of the KNHI Big Data, the number of people over 65 years old who were essential for participating in the program was 10,573 in the intervention region and 7648 in the control region. The study subjects comprised 4950 patients aged 65~84 years with hypertension in the intervention region and 3664 in the control region (Figure 1). 

### 2.3. 1:1 Matching for Coping with Confounders

To eliminate the influence of strong constitutional confounders like age and sex of study subjects [25], individual matching methods were used in this study. The variables used for individual matching were age, gender, type of subscription, and income rank as insurance premium level in patients with hypertension, as typical demographic variables, which are the eligibility criteria in the KNHI Big Data. 

The subscriber classification is divided into three types: self-employed, employee, and medical aid. It is further classified into two categories: household and household members (or dependent). Workers and employers, public officials, and faculty at all sites are registered employees. The self-employed refer to members other than insured employees and their dependents. The medical aid group refers to individuals with a difficult life according to the National Basic Living Security Act, Disaster Relief Act, and Act on Honorable Treatment of and Support for Persons who Died or were Injured for the Public Good. Household members (or dependents) are persons who primarily depend on the self-employed, employee, and medical aid [26]. Through 6 types of subscription, one can determine the income and lifestyle of the subjects. 

The premium category is a variable that divides the premium paid by the subscriber into 20 sub-categories. Generally, these are allocated between 1 and 20 sub-categories depending on their income. The higher the income, the higher the sub-category. However, if for the beneficiary of medical aid, the premium is not charged or charged after the event, they are entered as missing [19]. In this study, the beneficiary of medical aid had the premium sub-category as zero (0), and others were considered “missing.” The level of income can be indirectly determined through the sub-category of premiums. Therefore, we divided the five sub-groups for income rank. The I group of income rank is included in the 0th to 4th sub-categories; the II group, 5th~8th; the III group, 9th~12th; the IV group, 13th~16th; and the V group, 17th~20th. Each of the missing values and the 0~20th sub-categories were used for matching.

Based on these variables, the same patients were exactly matched in a 1:1 ratio. If two patients with the same variables were found in the intervention and control groups, they were matched. Moreover, if more than one case had the same variables, a random number of individuals within the group was given and matched. 

### 2.4. Investigating Indicators for Medication Adherence 

Based on the Andersen model [27], the community-based intervention is a chronic disease management program for addressing the enabling and need factors. It directly supports medical care and pharmaceutical costs, thereby increasing the possibility of medical service use for improving medication adherence. Moreover, health education and counseling services are increasing the awareness regarding medication adherence in a small way. Medication adherence may be measured indirectly or directly. Two indirect adherence metrics used in research and administrative settings are the medication possession ratio (MPR) and the proportion of days covered (PDC). These measures rely on pharmacy on insurance claims data. In clinical settings, adherence may be indirectly assessed using not only patient recall (self-report, questionnaire) but also other methods such as pill counting, dose counting device, and electronic prescribing, Direct methods, including observed therapy, and blood or urine drug and metabolite concentrations are most commonly used in research [28,29,30]. In this study, we modified the data used in the NHIS to suit the measurement used in research and administrative settings. Therefore, this study used the number of patients and proportion using insurance benefit days for the treatment of hypertension based on year and region, dispensation per prescription (DPP), and dispensation days per patient (DDPP) with hypertension as proxy indicators [31].

The number of patients and proportion by insurance benefit days for the treatment of hypertension by year indicates how many patients with hypertension have been continuously managed within the year. In chronic disease management program, the frequency of visits greatly varies depending on the patient’s long-term conditions and the nature of medical institutions. In order to determine whether a patient is continuously visiting for hypertension treatment, the total number of insurance benefit days per year was assessed. The insurance benefit days refers to the number of days on which the medication was prescribed to the patient on the date the patient visited the medical institution. Therefore, beneficiaries with ≥240 days were defined as patients with continuous treatment in this study, which is about two-thirds of the number of days covered by the NHIS within the year (i.e., 365 days.) The DPP indicator provides an indication of the patients’ adherence with the treatment in a medical institution. Moreover, the “DDPP with hypertension” indicator shows the average dispensation days per month for a number of patients with hypertension within a community, and the size of medical care was evaluated as an individual unit. 

### 2.5. Statistical Analysis

Firstly, in order to determine the homogeneity of study subjects between the intervention and control regions, the descriptive analysis including the t-test and chi-square test were used. 

Secondly, we established the dummy variable (time) before and after introducing the program and the dummy variable (region) that distinguish the districts regarding whether the program was introduced or not to apply the difference-in-difference (DID) regression methodology. In this study, if the district was an intervention group, region was considered 1, and if the district was a control group, region was taken to be 0 (zero). Moreover, if the study period was from January 2010 to December 2011, time was 0 (zero), and if the study period was from January 2013 to December 2014, time was 1. The DID regression analysis excluded the one-year period in 2012. This is because the period for normalization after implementing the program was considered 6 months. As it excluded 6 months after the program, it also excluded 6 months before the program in order to match the pre- and post-intervention periods. After creating the “DID” variable by multiplying region and time, a regression analysis was performed [32].
$$\begin{array}{l} y=intercept+\beta_{1}region+\beta_{2}time+\beta_{3}DID\\ region\{\begin{array}{c} 1=Hongcheon\\ 0=Hoengseong \end{array},\;time\{\begin{array}{c} 1=201301\sim 201412\\ 0=201001\sim 201112 \end{array},\\ DID=region\times time \end{array}$$

We set the indicator that can represent the effects as a dependent variable (*y*). Using these, we performed a regression analysis to estimate the intercept and coefficients (*β*_1_, *β*_2_, *β*_3_). The intercept is the average value (*y*) of a region that did not implement the program before it was introduced. *β*_1_ is the difference of the average value (*y*) between the region that either implemented or did not implement the community-based intervention before it was introduced. Therefore, the difference between regions may be determined before the introduction of the program. *β*_2_ is the difference of the average values (*y*) between pre-intervention and post-intervention periods in a region where the community-based intervention is not implemented. Therefore, it corresponds to the natural increase in *y*. *β*_3_ is the difference in the average value (*y*) before and after implementing the intervention in the region where the program was introduced, minus the difference in the average value (*y*) before and after implementing the intervention in the region where the program was not introduced.

The data of this study was extracted and provided by the KNHI Big Data in accordance with the study purpose and did not include the identification of patients’ personal unique information. For the ethical consideration of this study, the review exemption was also approved by the Institutional Review Board of Yonsei University Wonju Severance Christian Hospital (approval no. CR317342). 

## 3. Results

### 3.1. Comparison of Demographic Characteristics for Final Study Subjects Pre-Matching and Post-Matching between Study Regions

Table 1 shows the result of the homogeneity test for the final study subjects pre-matching and post-matching between the intervention and control regions. According to the basic eligibility information before matching, the 1:1 match was done so that the final study subjects for the intervention and control region were similar, from the gender-to-age ratio to the level of income. The final number of subjects for the 1:1 match was 2685 in the intervention region and 2685 in the control region.

For homogeneity tests, age, which is a continuous variable, was tested using the t-test, whereas the categorical variables, gender, type of subscription, and income rank, were evaluated using the chi-square test. Prior to matching, the study subjects between the intervention and control regions had almost similar distributions of gender ratio and income ranks; however, the distributions for age and type of subscription yielded statistically significant differences between the two regions. After matching, the final study subjects between the intervention and control regions could be observed to have the same demographic characteristics, from the perspectives of predisposing factors on the Andersen model.

Of course, since they were hypertensive patients, they all need chronic disease management services to prevent complications of severe CVD.

### 3.2. Comparison of Healthcare Resources between Study Regions

Table 2 shows the characteristics of healthcare resource distribution between the intervention and control regions. Across all types, the medical institutions in the intervention region were slightly larger than those in the control region. In both regions, the number of medical institutions was higher in 2014 compared to 2010. The institutions closely related to the community-based intervention are clinics for internal medicine class and pharmacies. Comparing the population-to-healthcare resources, healthcare resource indicators in the intervention region were slightly higher than those in the control region, but with no significant difference. Healthcare resource indicators based on pharmacies were very similar in both regions. From the perspectives of enabling factors in the Andersen model, the distribution of healthcare resources in the intervention and control regions was similar. 

### 3.3. Comparison of Medication Adherence for Assessing Community-Based Intervention

Table 3 shows the trends in the number of patients by insurance benefit days for the treatment of hypertension between the intervention and control regions. Finally, 2685 subjects through matching were selected at the baseline. Among them, 2174 patients received continuous insurance benefits for ≥240 days in 2014 in the intervention region. In other words, it was about 81.0% of the total number of patients. On the other hand, 2138 patients were included in the control region, comprising about 79.7% of the total number of patients. We could observe that the number of patients with continuous treatment in the intervention region was higher than that in the control region. On the contrary, those with zero insurance benefit days in 2014 were 221 in the intervention region, about 8.2% of the total number of patients. In addition, in the control region, 229 patients had zero insurance benefit days in 2014, about 8.5% of the total number of patients. In the control region, the number of patients with zero insurance benefit days was slightly higher than that in the intervention region. 

Table 4 shows the result of the difference-in-difference regression analysis based on the intervention and control regions, and the pre-intervention and post-intervention periods, using the proxy indicators for medication adherence for treating hypertension. After implementing the community-based intervention for chronic diseases management program, the cases of DPP in the intervention region increased approximately 10.88% on average per month over the control region (Figure 2). Moreover, the DDPP with hypertension in the intervention region increased by 2.2 days on average per month over the control region (Figure 2). In other words, patients with hypertension in the intervention region had been more on medication for about 26.4 days (2.2 days × 12 months) during one year than those in the control region. 

## 4. Discussion

### 4.1. Comparison with Previous Studies 

Continuous treatment of patients with hypertension is important, including a regular visit (prescription) to healthcare institutions and taking medicine regularly (patients’ adherence) due to the purchase of antihypertensive drugs (dispensation) at pharmacies [15,16]. Some international studies similar to ours have been conducted in Brazil, Taiwan, the USA, China, Indonesia, England, and Japan [33,34,35,36,37,38,39]. Overall, these community-based healthcare programs were an effective means of helping patients’ adherence with hypertension. A meta-analysis of community-based interventions, to enhance medication adherence and blood pressure control in hypertension, documented significant but modest post-intervention improvements in blood pressure outcomes among hypertensive patients [40]. Even more, supported self-management can improve blood pressure control according to another systematic qualitative and quantitative meta-review [41]. 

Medication nonadherence is widespread and varied by disease, patient characteristics, and insurance coverage, with nonadherence rates ranging from 25% to 50% [42,43]. Approximately 50% of patients with cardiovascular disease have poor adherence to their prescribed medications [44]. In order to improve medication adherence, clinicians as well as health policy decision makers must understand why patients fail to take prescribed medications, that is, the determinants of medication nonadherence. These determinants can usually be categorized as patient related, provider related, and external factors [30]. Among these factors, this community-based intervention is mainly conducted following the external factors in terms of strengthening the health care system according to the chronic care model [12]. The expected effects of this intervention program include also improved awareness and self-management rate of the target disease (hypertension) through health education and counseling for registered patients and extended healthy lifespan by prevention of complications from cardio-cerebrovascular diseases through continuous management of hypertension [13,40,41,45]. Especially our study applied to the Andersen’s medical use model as the expected effects of the program, confirming continuous medication adherence [27]. In order to investigate the purpose of our study, the number of patients by insurance benefit days for managing hypertension, proportion of DPP, and DDPP with hypertension were selected as proxy indicators. The trend of patients by insurance benefit days between study regions can be determined by the continuous management level for hypertensive patients within one year. Under KNHI administrative settings, medication adherence can be indirectly obtained through DPP, and the actual effect can be estimated through the DDPP with hypertension. All indicators in Hongcheon County have significantly increased since 2012, the year in which the community-based intervention was implemented. This confirmed that continuous medication adherence has significantly increased since the program implementation, even after using the difference-in-difference regression analysis [32]. 

In this rapidly aging society, the incidence rates of hypertension significantly increased with age [24,46,47]. During the five-year follow-up, 26.7% of non-hypertensive individuals developed incident hypertension [46]. Thus, the community-based hypertension control programs have been conducted for more than 10 years in South Korea [13,45,48]. Among them, according to a study that used big data from the NHIS to evaluate the effectiveness of the chronic disease management program in Incheon, the prescription days of outpatients, continuous treatment rate, days of inpatient care, days of outpatient visits, and cost of inpatient and outpatient care in the registered group were higher compared to the non-registered group. Unlike our study; however, the shortcomings of that study were as follows: The research period was relatively short as one year (365 days) before and after intervention; the criteria for selecting the control group were ambiguous; and the intervention and control groups did not match [48]. 

After the 2012 UN Declaration on the Prevention and Control of NCDs [1], the World Health Organization and the World Heart Federation adopted the global goals “25by25”, to reduce early death rates by 25% due to NCDs by the year 2025. In addition, they proposed detailed strategies such as reducing the rate of increase in blood pressure by 25% and maintaining 0% as the rate of diabetes/obesity [45,49,50,51]. In order to achieve the Global NCDs Action Plan 2013–2020 in conjunction with the UN’s Sustainable Development Goals (SDG) in 2015, investing in NCDs prevention and management (SDG 3.4) is strongly recommended to all countries [52]. Therefore, the Korean government has come to a time for expanding and strengthening from the current pilot stage to the nationwide level through the comprehensive reassessment of the community-based intervention for hypertension control. 

### 4.2. Methodological Consideration 

Thus, in order to evaluate the effectiveness of the community-based intervention in this study, not only a well-designed research methodology tracking longitudinally but also a sophisticated theoretical model should be considered. To minimize the unexpected bias caused by potential confounding variables [25], the eligibility information of KNHI Big Data in our study are refined as follows: First, a control region similar to all characteristics of the intervention region was selected, except the community-based intervention for controlling hypertension. Second, this program is a community-based hypertension intervention that approaches the enabling and need factors according to the Andersen’s model [27]. In terms of enabling factors, the center supports elderly patients aged ≥65 years to promote continuous medication adherence. Moreover, the center educates and manages patients so that they become aware of hypertension in terms of need factors. Direct interventions that enable patients to receive continuous medication adherence can be observed as support out-of-pocket payment for medical care and medication fee. Thus, as defined in the community-based intervention, the study subjects determined that elderly patients aged ≥65 years and <85 years were only included in the program. Third, the size and characteristics of study subjects between the intervention and control regions were equally matched, 1:1, by establishing a cohort by a non-equivalent control group design in terms of predisposing factors. 

Some limitations of this study were identified due to the weakness of the study data. First, we could indirectly check the patients’ adherence by this intervention until the dispensation after the prescription using secondary insurance claims data; however, we could not actually determine whether patients’ blood pressure was controlled by anti-hypertensive medications. Second, patients with hypertension in 2010 were not included as the study subjects unless they visited the medical institution. Third, there was no confirmation of complications due to hypertension and the medical expenses incurred. Medication nonadherence leads to poor outcomes, which then increase health care service utilization and overall health care costs. Between US$ 100 and US$ 300 billion of avoidable health care costs have been attributed to nonadherence in the US annually, representing 3% to 10% of total US health care costs [30,53]. Lastly, we could not apply the Patient Assessment of Chronic Illness Care instrument [54], including indirectly administrative measurements (MPR, PDC) [28,30] as well as Korean-version eight-item Morisky Medication Adherence Scale (MMAS-K) [55,56] for measuring the extent to which hypertensive patients receive care congruent with the chronic care model. 

## 5. Conclusions

In this study, we did a comparative analysis regarding the gap in medication adherence of hypertensive elderly patients based on Andersen’s model between Hongcheon County, with the hypertension and diabetes registration and education center, and Hoengseong County, without this center. For the aim of this study, the change in the medication adherence in elderly patients with hypertension before and after implementing the community-based intervention between two regions was verified through descriptive analysis and the difference-in-difference regression analysis using the KNHI Big Data. Summarily, elderly patients with hypertension in Hongcheon County have been receiving continuous management and improving medication adherence compared to those in Hoengseong County, in 2013 and 2014, since this program was implemented. In conclusion, the findings suggest that the community-based intervention for controlling hypertension should expand nationwide in Korea, ahead of a super-aged society. Of course, we need additional information about a delay in the period of developing complications of hypertension within the intervention region. Additionally, we are planning to provide a cost–benefit estimate through rolling out this community-based intervention nationwide in the near future. 

## Figures and Tables

**Figure 1 ijerph-16-00721-f001:**
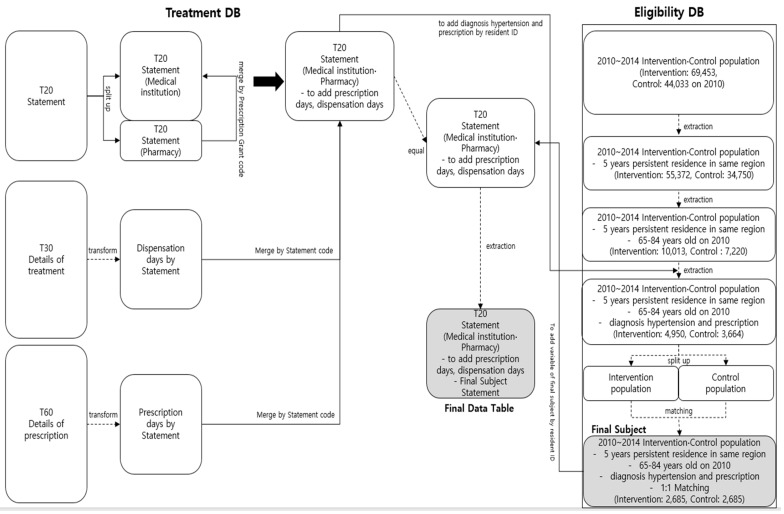
Flowchart for selecting study subjects using the Korean National Health Insurance (KNHI) Big Data on the community-based intervention program.

**Figure 2 ijerph-16-00721-f002:**
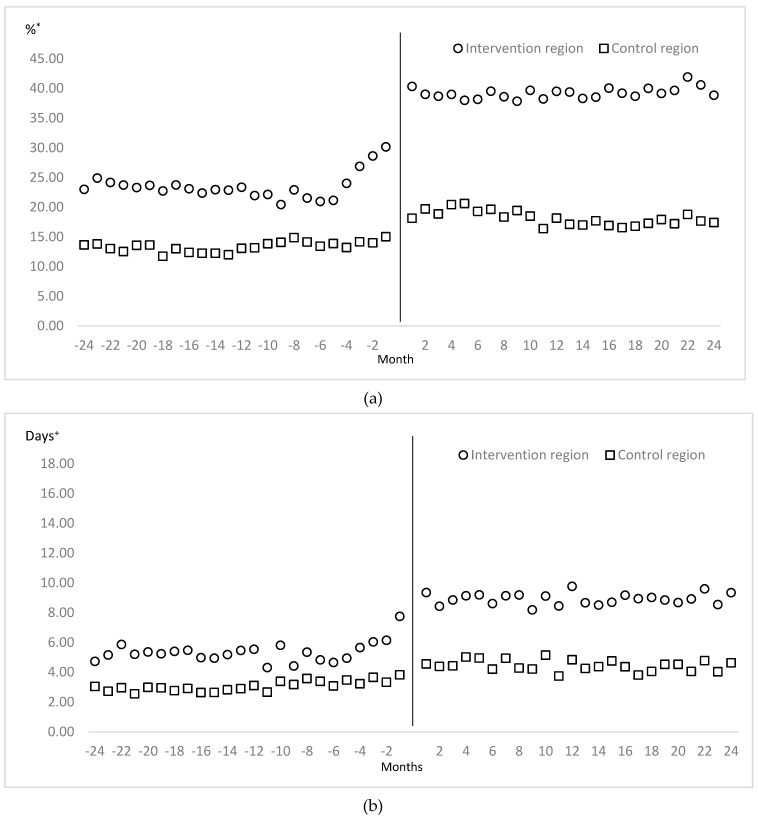
Comparison of dispensation per prescription (**a**) and dispensation days per patient with hypertension (**b**) on pre-intervention and post-intervention between study regions. ***** Proportion of dispensation per prescription on hypertensive patients during given period. + Dispensed days per patient with hypertension during given period.

**Table 1 ijerph-16-00721-t001:** Results of homogeneity test on final study subjects according to matching between study regions: Predisposing and need factors on the Andersen model.

Pre-Matching						Post-Matching				
Continuous Variable										
	Intervention Region		Control Region		*p*	Intervention Region		Control Region		*p*
	n_1_	Mean (SD)	n_2_	Mean (SD)		n_3_	Mean (SD)	n_4_	Mean (SD)	
Age (years)		72.33		72.71			72.38		72.38	
	4950	(4.91)	3664	(4.96)	0.0120	2685	(4.74)	2685	(4.74)	1
Categorical variables										
	n	%	n	%		n	%	n	%	
Gender										
Male	1846	37.29	1385	37.80	0.6307	934	34.79	934	34.79	1
Female	3104	62.71	2279	62.20		1751	65.21	1751	65.21	
Type of subscription										
Self-employed	1156	23.35	749	20.44	0.0015	510	18.99	510	18.99	1
Self-employed dependents	695	14.04	457	12.47		296	11.02	296	11.02	
Employee	115	2.32	99	2.70		32	1.19	32	1.19	
Employee dependents	2597	52.46	2034	55.51		1585	59.03	1585	59.03	
Medical aid households	309	6.24	258	7.04		209	7.78	209	7.78	
Medical aid dependents	78	1.58	67	1.83		53	1.97	53	1.97	
Income rank *										
I	963	20.13	768	21.41	0.3336	535	20.40	535	20.40	1
II	481	10.06	375	10.45		208	7.93	208	7.93	
III	675	14.11	511	14.25		330	12.59	330	12.59	
IV	1163	24.32	810	22.58		626	23.87	626	23.87	
V	1501	31.38	1,123	31.31		923	35.20	923	35.20	

Figures are frequencies (column percent). N, number. * Total may not match due to missing values

**Table 2 ijerph-16-00721-t002:** Comparison of healthcare resources between study regions: Enabling factors on the Andersen model.

	Intervention Region					Control Region				
	2010	2011	2012	2013	2014	2010	2011	2012	2013	2014
Healthcare Institutions										
Clinic	29	27	28	28	29	13	13	12	13	12
Health center	1	1	1	1	1	1	1	1	1	1
Health sub-center	8	8	8	8	8	8	8	8	8	8
Health post	18	18	18	18	18	8	8	8	8	8
Pharmacy	25	25	24	24	26	15	15	14	15	17
Total	81	79	79	79	82	45	45	43	45	46
Resource * per 1000 population ^+^ (on clinic)										
No. of clinic per 1000 population	0.42	0.39	0.40	0.40	0.41	0.30	0.30	0.27	0.29	0.27
No. of bed per 1000 population	3.15	3.12	2.95	2.94	2.89	1.09	0.68	0.68	0.67	0.67
No. of physician per 1000 population	0.52	0.49	0.49	0.50	0.53	0.34	0.34	0.29	0.31	0.29
No. of nurse per 1000 population	0.29	0.29	0.30	0.27	0.26	0.05	0.02	0.02	0.02	0.02
Resource * per 1000 population (on pharmacy)										
No. of pharmacy per 1000 population	0.36	0.36	0.35	0.34	0.37	0.34	0.34	0.32	0.34	0.38
No. of pharmacist per 1000 population	0.48	0.45	0.43	0.46	0.49	0.36	0.36	0.32	0.34	0.38

* The Korean National Health Information Database among Korean National Health Insurance (KNHI) Big Data; ^+^ Resident population by year and region

**Table 3 ijerph-16-00721-t003:** Comparison of insurance benefit days * for treating hypertension between study regions.

Region	Intervention Region					Control Region				
Year	2010	2011	2012	2013	2014	2010	2011	2012	2013	2014
Study subjects	2685	2685	2685	2685	2685	2685	2685	2685	2685	2685
	(100.0)	(100.0)	(100.0)	(100.0)	(100.0)	(100.0)	(100.0)	(100.0)	(100.0)	(100.0)
0	0	152	162	194	221	0	134	150	192	229
	(0.0)	(5.7)	(6.0)	(7.2)	(8.2)	(0.0)	(5.0)	(5.6)	(7.2)	(8.5)
1~179	311	178	160	150	189	311	175	176	179	224
	(11.6)	(6.6)	(6.0)	(5.6)	(7.0)	(11.6)	(6.5)	(6.6)	(6.7)	(8.3)
180~239	146	112	104	83	101	143	121	122	98	94
	(5.4)	(4.2)	(3.9)	(3.1)	(3.8)	(5.3)	(4.5)	(4.5)	(3.7)	(3.5)
240~359	1098	1012	1056	963	894	1101	1127	1043	1005	885
	(40.9)	(37.7)	(39.3)	(35.9)	(33.3)	(41.0)	(42.0)	(38.9)	(37.4)	(33.0)
≥360	1130	1231	1203	1295	1280	1130	1128	1194	1211	1253
	(42.1)	(45.9)	(44.8)	(48.2)	(47.7)	(42.1)	(42.0)	(44.5)	(45.1)	(46.7)

Figures are frequencies (column percent). * Total percentage may not match to 100 due to rounding.

**Table 4 ijerph-16-00721-t004:** The result of difference-in-difference regression analysis using the indicators for medication adherence to patients with hypertension.

Indicator	Variable	*β*	*t*	*p*-Value
Dispensation per prescription (DPP)	Intercept	13.36	45.68	<0.0001
	Region	10.16	24.56	<0.0001
	Time	4.80	11.60	<0.0001
	DID	10.88	18.59	<0.0001
Dispensation days per patient (DDPP) with hypertension	Intercept	3.07	31.62	<0.0001
	Region	2.28	16.62	<0.0001
	Time	1.38	10.06	<0.0001
	DID	2.20	11.34	<0.0001

DID: ‘Difference-in difference’ variable by multiplying region and time according to equation.

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
