# Peer review of "A Community-Based Intervention for Improving Medication Adherence for Elderly Patients with Hypertension in Korea"

_ijerph, 2019, doi:10.3390/ijerph16050721_

Round 1

Reviewer 1 Report

The authors conclude "the findings suggest that the community-based intervention for chronic disease management should expand nationwide in Korea". 

Can they make an estimate as to the total cost of rolling out this community-based intervention nationwide? Can they provide a cost-benefit estimate (in the case that the costs can be offset by the financial benefits of enhanced adherence to hypertensive medications> 

Author Response

Comments and Suggestions for Authors:

Can they make an estimate as to the total cost of rolling out this community-based intervention nationwide? Can they provide a cost-benefit estimate (in the case that the costs can be offset by the financial benefits of enhanced adherence to hypertensive medications> 

Our answer:

Because the cost-benefit estimate is another research project, my team are preparing separately.

Therefore, I add some information about avoidable health care costs according to medication nonadherence in the US (lines 414 to 417).

Also, in conclusion, these below sentences were inserted: "we are planning to provide a cost-benefit estimate through rolling out this community-based intervention nationwide in the near future" (lines 437 to 439).

Reviewer 2 Report

Review of Int J of Env Res Pub Health Korea and adherence

Abstract

To me the Abstract needs to be re-structured in order to provide some background before setting out the aim.

This matters for example as the second last sentence sets out what the intervention  does.

Introduction

The framework for the study is Andersen’s medical model (ref 18). Yet I had no idea what that was and am a little surprised that this very high level model was used. Regardless, a search for the model reveals that there are more current versions, and these should be mentioned. Indeed, given the importance of a framework, I would have expected some diagram to show me what is expected of the variables included in the model and the relationships between them. At least more explanation.

Throughout the manuscript the authors mention medication compliance yet this is a fairly dated concept, and particularly in an intervention designed to improve the rate of medication-taking, it would be wise to consider using the term adherence and stick to this term, unless there is a good reason not to. The terms are similar but medication adherence is much more specific and patient-centered.

There is scant information about the intervention. What does the intervention actually do? Is it just a reminder service (line 66) No, but the next sentence is meant to tell us what it is, presumably. But if you wish medical and pharmacy people to read the article, you need more detail about this. Surely, there is some qual of quant research to tell us what is involved? Not just a reference?

Methods.

Throughout the methods results and discussion, you mention the erm dispensation rates as a measure of adherence and give a reference for this (the Cochrane article). However, the Cochrane article does not use this term. Methodologically, you need to set out a case for why you use dispensation rates, with references.  Other related concepts include Mean Possession Ration for example, which the calculations can be shown.

Results:

The methods and results seem appropriatey presented.

Figure2 requires labels beside % and Days

Discussion

Normally, the first paragraph of a Discussion sets out the main findings. Yet, the first mention of your own findings till line 315, after wading through what should be in the Introduction.

The sentence on Line 309 “confirmed continuous compliance is mystifying’. What is continuous medication compliance?

Ok, so intervention seems to work, How much, practically, does it work. What is a great result? Is a 10% rise in dispensations per month good? Is 26.4 days per year a good? How does this compare to other interventions? Does this rise justify the intervention? Is the rise big enough. A little context please, before stating “Should expand nationwide in Korea…” (Line 376). What does a 10% rise in adherence do to BP lowering? To mortality rates?  

Author Response

Abstract

These sentences about background were inserted: "The chronic disease management program, a community-based intervention including patient education, recall & remind service, and reduction of out-of-pocket payment, was implemented in 2005 in Korea to improve patients’ adherence for antihypertensive medications (lines 17 to 19).

Introduction

1) Informations about the intervention  were added more in lines 52 to 55 and 75 to 78.

2) The definition about medication adherence were inserted (lines 83 to 84).

       3) Mention of Andersen’s medical model were deleted in this section (lines 101). Because it's not the main frame, it's the model for analysis (line 183 to 184). So, this application were added in the discussion (lines 355 to 357).

   3. Methods

    Mention about medication adherence were inserted (lines 187 to 195).

  4. Results

   labels in Figure 2 were added (lines 320 to 327).

   5. Discussion

     1) Significant meaning of intervention were inserted (lines 330 to 362).

     2) Limitaions about measurements for medication (non)adherence using insurance claims data were added (lines 408 to 421).

    3) More information and future research direction related this study were added (line 435 to 439).

Reviewer 3 Report

Interesting study and sound research design. Just couple comments: 1) Section 1.1 needs some more editing. For example, lines 51 to 53 need to be rephrased to be understood correctly. Also, the sentences about the program needs to be edited. 2) I am not sure whether Tables 1 and 3 are necessary. Since that information is already in the text, they can be avoided to reduce the length of the manuscript.

Author Response

1) More explanation including some references in Section 1.1 were added (lines 52 to 78).

2) Table 1 was deleted. Instead, the explanation about Table 1 was put together in Section 2.2 (lines 141 to 151).
